# PARIS: Planning Algorithms for Reconfiguring Independent Sets

**Remo Christen,**[1] **Salomé Eriksson,**[1] **Michael Katz,**[2] **Christian Muise,**[3]
**Alice Petrov,**[3] **Florian Pommerening,**[1] **Jendrik Seipp,**[4] **Silvan Sievers,**[1] **David Speck**[4]

[1] University of Basel, [2] IBM T.J. Watson Research Center, [3] Queen's University, [4] Linköping University
{remo.christen, salome.eriksson, florian.pommerening, silvan.sievers}@unibas.ch, michael.katz1@ibm.com,
{christian.muise, 17ap87}@queensu.ca, {jendrik.seipp, david.speck}@liu.se

## Abstract

Combinatorial reconfiguration studies how one solution of a combinatorial problem can be transformed into another. The transformation can only make small local changes and may not leave the solution space. An important example is the independent set reconfiguration (ISR) problem, where an independent set of a graph (a subset of its vertices without edges between them) has to be transformed into another one by a sequence of modifications that remove a vertex or add another that is not adjacent to any vertex in the set. The 1st Combinatorial Reconfiguration Challenge (CoRe Challenge 2022) was a competition focused on the ISR problem. Our team participated with two solvers that model the ISR problem as a planning problem and employ different planning techniques for solving it. They successfully competed in the challenge and were awarded 4 first, 3 second, and 3 third places across 9 tracks. In this work, we show how to model ISR problems as planning tasks and describe the planning techniques used in our solvers. For a fair comparison of ISR approaches, we re-run the entire competition under equal computational conditions. Besides showcasing the success of planning technology, we hope that this work will create a cross-fertilization of the two research fields.

## 1 Introduction

Combinatorial reconfiguration studies the space of solutions for combinatorial problems. The task is to transform one solution of a combinatorial problem into a different one, without leaving the space of solutions. Each transformation can only make a small local change to the current solution. The term was coined by Ito et al. (2008) who show that there is a host of problems derived from NP-complete (combinatorial) problems that fall into the category of combinatorial reconfiguration problems and that they are PSPACE-complete. Two prominent examples for reconfiguration tasks are propositional satisfiability (Ito et al. 2011) and graph $k$-coloring (Cereceda 2007). But probably the most well-studied representative of combinatorial reconfiguration tasks is the *independent set reconfiguration* (ISR) problem (Kaminski, Medvedev, and Milanic 2012).

An independent set of a graph is a subset of its vertices such that no two vertices of the subset share an edge. Reconfiguring an independent set means replacing one vertex in the subset with another one such that the new subset is still an independent set. The ISR problem is to find a sequence of such reconfiguration steps to reach a given target configuration from a given start configuration. The problem is PSPACE-complete (Nishimura 2018), which means it is as hard as automated planning (Bylander 1994).

The *1st Combinatorial Reconfiguration Challenge* (CoRe 2022)[1] is a competition that compares practical combinatorial reconfiguration algorithms. Its first instantiation targeted the ISR problem, featuring different tracks. We participated in the competition using two solvers that model ISR problems as planning tasks and use various planning techniques for solving them. Among the seven teams that participated, our solvers achieved 4 first, 3 second, and 3 third places across all tracks, winning the majority of awards.

In this work, we present the ISR problem and explain how we can model it as a planning problem. We describe the technology used in our solvers, which is mostly based on planning techniques, including a technique for detecting unsolvable problems which we believe to be useful for unsolvability planning in general. Furthermore, since competitors of the competition ran their solvers themselves using different hardware and resource limits, we re-ran all of them under equal computational conditions and report the results in this work. Besides showcasing the success of planning technology, we also introduce a problem that is new to our community. We believe this will lead more planning researchers to develop ideas for the ISR problem and create a cross-fertilization of the fields.

## 2 Background

A *graph* is a pair $G = \langle V, E \rangle$, where $V$ is a set of *vertices* and $E \subseteq \{\{u, v\} \mid u, v \in V, v \neq u\}$ is a set of *edges* between the vertices. An *independent (vertex) set* of a graph $G$ is a subset of vertices $I \subseteq V$ such that no two vertices in the subset $I$ are edges of $G$, i.e., for all $v, u \in I$ it holds that $\{v, u\} \notin E$.

### 2.1 Independent Set Reconfiguration

Similar to Kaminski, Medvedev, and Milanic (2012), we consider an independent set as a set of tokens placed on the vertices of a graph $G$, called *token configuration*, such

---

[1]https://core-challenge.github.io/2022

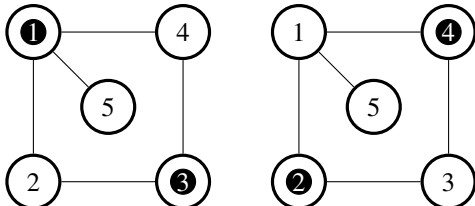

Figure 1: Visualization of the independent set reconfiguration problem described in Example 1 with a graph consisting of five nodes, two tokens depicted in black, the start independent set $I_s$ (left), and the target independent set $I_t$ (right).

that no two tokens are adjacent. The *token jump* reconfiguration rule describes how to transform one token configuration into another, moving a token from one vertex to any other unoccupied vertex, so that the resulting configuration again describes an independent set. Note that the token can jump, i.e., it does not have to move along an edge. Given the reconfiguration rule, we define a reconfiguration sequence $\rho = \langle I_0, \ldots, I_n \rangle$ as a sequence of non-repeating independent sets, where each set $I_i$ with $1 \leq i \leq n$ results from a single token jump from the previous set $I_{i-1}$. The length $|\rho|$ of a reconfiguration sequence $\rho = \langle I_0, \ldots, I_n \rangle$ is the number of token jumps inducing the sequence, i.e., $|\rho| = n$. The Independent Set Reconfiguration decision problem (Kaminski, Medvedev, and Milanic 2012) is defined as follows.

**Definition 1 (Independent Set Reconfiguration)** *Given a graph $G$ and two independent sets $I_s$ and $I_t$, the* independent set reconfiguration (ISR) decision problem *is to determine whether there exists a reconfiguration sequence $\rho = \langle I_s, \ldots, I_t \rangle$.*

The ISR problem is one of the most prominent representatives of combinatorial reconfiguration. It is known to be PSPACE-complete for general input graphs (Kaminski, Medvedev, and Milanic 2012; Nishimura 2018) and formed the central problem of CoRe 2022.

**Example 1** *Figure 1 shows an ISR problem with the start set $I_s = \{1, 3\}$ and the target set $I_t = \{2, 4\}$. A solution to this problem is the reconfiguration sequence $\rho = \langle \{1, 3\}, \{3, 5\}, \{2, 5\}, \{2, 4\} \rangle$, where first the token at node 1 is moved to node 5, then the token at node 3 is moved to node 2, and finally the token at node 5 is moved to node 4. This sequence has a length of $|\rho| = 3$, since it performs three jumps and is the shortest sequence that solves the problem.*

## 2.2 Combinatorial Reconfiguration Challenge

Similar to the International Planning Competition (IPC), CoRe 2022 featured different tracks. They can be separated into two main categories: graph tracks and solver tracks.

**Graph Tracks** In the *graph* tracks the objective was to construct an ISR instance such that the shortest reconfiguration sequence is as long as possible. For CoRe 2022 there were three graph tracks, one each for graphs with 10, 50 and 100 nodes, and the team that constructed the instance with the longest shortest reconfiguration sequence won the respective track.

**Solver Tracks** In total, there were three different solver tracks in CoRe 2022: the *existent*, the *shortest* and the *longest* track, each further subdivided into a *single solver* sub-track and a *portfolio solver* sub-track. In the *existent* track, each solver that provided a reconfiguration sequence for or detected unsolvability of an ISR instance received one point. In contrast, the *shortest* and *longest* tracks considered the quality of the solutions, and solvers that provided the shortest/longest (among the participants) reconfiguration sequence for an instance received one point.[2] The winning solver for each track was the one that received the most points across all benchmark ISR instances.

Note that the names *shortest* and *longest* are somewhat misleading. The aim in these tracks is to find a solution, aiming at as short/long loopless solutions as possible, but no guarantees on optimality are needed. To draw parallels between these tracks and International Planning Competition (IPC) tracks, the *shortest* track is actually more similar in that respect to the *satisficing* IPC track. Currently, there is no equivalent in planning competitions to the *longest* track. The *existent* track is somewhat similar to the *agile* IPC track.

## 2.3 Classical Planning

In this paper, we propose to model the ISR problem as a classical planning problem. For this, we consider the *Planning Domain Definition Language* (PDDL) (McDermott et al. 1998) and the $SAS^+$ formalism (Bäckström and Nebel 1995) to describe classical planning problems. A *(classical) planning problem* is a concise representation of a transition system with a single initial state, a compact description of the set of goal states, and a set of actions with preconditions and effects that describe the transitions. The objective is to derive a course of action that transforms the initial state into one of the goal states. While the full details of PDDL are beyond the scope of this paper and are not necessary to follow the content of the paper, the excerpts presented in this paper suffice to present our contributions. For a more detailed account, we refer the reader to Haslum et al. (2019).

An $SAS^+$ task formally is a tuple $\langle \mathcal{V}, \mathcal{A}, \mathcal{I}, \mathcal{G} \rangle$, where $\mathcal{V}$ is a finite set of *variables* $V$, each with a finite domain $dom(V)$, $\mathcal{A}$ is a finite set of *actions*, $\mathcal{I}$ is the *initial state*, and $\mathcal{G}$ is the *goal*. Partial variable assignments $p$ map a subset of variables $vars(p) \subseteq \mathcal{V}$ to values in their domain. Variable assignments $s$ with $vars(s) = \mathcal{V}$ are called *states*. A partial variable assignment $p$ is satisfied in a state $s$ if $p$ and $s$ agree on $vars(p)$. Each action $a \in \mathcal{A}$ consists of a precondition $pre(a)$ and an effect $eff(a)$, both partial variable assignments. An action is applicable in a state if its precondition is satisfied, and applying it updates the state with values defined in its effect. A planner finds a sequence of actions that is sequentially applicable and leads from the initial state $\mathcal{I}$ to some state satisfying the partial variable assignment $\mathcal{G}$.

# 3 Graph Track

The graph track was dedicated to finding challenging ISR instances. Our entry finished tied for third in the $n = 10$ in-

---

[2]Reconfiguration sequences must be non-repeating. Therefore, participants must search for loopless solutions in the longest track.

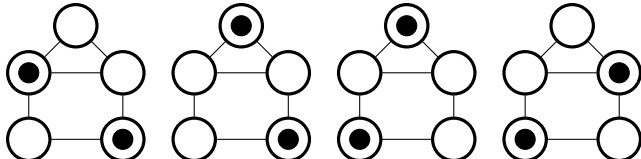

Figure 2: Reconfiguration sequence from *off* to *on*.

stance, and second place for the $n \in \{50, 100\}$ instances. Drawing from the notion of "gadgets" in computational complexity, we leverage a five-node subgraph called the "house widget" in order to encode bit flips in a graph and thus require an exponential plan length (exponential in the number of widgets included). Each subgraph consists of a 4-cycle with two adjacent nodes leading to a 5th node called the *anchor*. Figure 2 shows this widget and all of its maximally independent sets. We call the configuration on the left *off* and the configuration on the right *on*. The sequence in Figure 2 is the (only) way for a house widget to "flip its bit".

The house widget has a number of properties that make it ideal to use as a building block in creating exponential sequences: (1) the graph has an optimal "long" shortest reconfiguration sequence to flip its bit for ISR instances with 5 nodes, and (2) each step of the reconfiguration sequence consists of a maximum independent set. Also, (3) this reconfiguration sequence is unique and (4) the anchor is occupied throughout the entire sequence of flipping the widget with the exception of the starting state and ending state. Finally, (5) the solution space is a path, and thus the behaviour of the widget is predictable.

We treat our house widget as an individual bit and connect several of them in a way that ensures exponential solutions. First, we make the anchors a fully connected subgraph, guaranteeing that no two houses can switch states simultaneously. The order of bit flips is then enforced by connecting a house's anchor to the bits previously seen in the sequence. Figure 3 shows the connection that allows house 1 to switch only when house 2 is set to *on*. We add these connections in an iterative way, with the addition of every new house widget. As our base case, the first house has an initial configuration of *off* and a goal of *on*. Suppose we have a sequence of $k$ houses generated; we add house $k+1$ according to the following:

1. We can only flip house $k + 1$ when the goal of house $k$ is satisfied.
2. The new initial state is to have all houses *off*, and the new goal is to have only the house $k + 1$ *on*.

This forces the plan length to double with each new house: achieve the old goal of the $k$-house sequence, flip house $k + 1$, and then go back to the initial state of the $k$ house sequence. Thus, we have a set of subgraphs connected in such a way that forces exponential growth in plan length with every added house subgraph. Putting everything together, graphs are generated as follows:

1. Create $k$ houses.
2. Make the anchors of all houses a fully connected subgraph of order $k$.

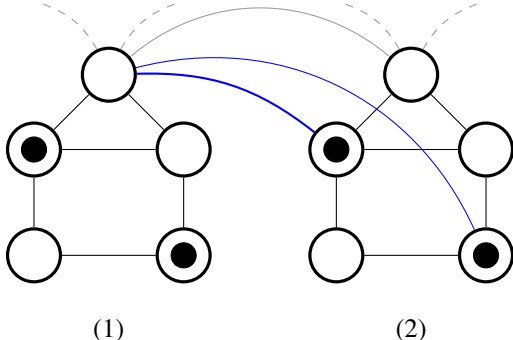

(1)                   (2)

Figure 3: House 1 is unable to flip unless house 2 is *on*.

3. Add edges according to the iterative method above.

## 3.1 Building on Planning Technology

What we present above is the culmination of extensive exploration and intuition-building on the problem of generating difficult planning instances for the ISR domain. Even though the final solution was free of planning technology, the journey to a competitive entry was ripe with planning-based insight. In the following, we highlight some of the planning elements that proved essential to this exploration.

Similar to the early exploration in the solver tracks, the PLANUTILS library (Muise et al. 2022) allowed us to explore the instances we were generating easily with different planners. Beyond that, we gained insight by leveraging state-of-the-art planners to compute the worst-case goal configurations. To evaluate solutions for the graph track, we used a manually modified version of Fast Downward's A* search with the blind heuristic, effectively giving us breadth-first search (BrFS), in the following way:

1. Compute a target state with the furthest distance from the initial state, by exhaustively expanding the search space with BrFS (we modified the planner to output one such target state).

2. Use this newly-found state as a new initial configuration.

3. Repeat step 1 to find a new candidate initial state.

For some solution attempts, the above approach allowed us to find not only a reasonable goal choice but an initial state candidate as well. Finding pairs of states that are maximally far apart is closely related to the computation of upper bounds for factored state-space search (Abdulaziz, Gretton, and Norrish 2017), a problem that is NEXPTIME-hard.

Finally, the generated problems are meant to yield very long shortest plans. We were able to use the breadth-first search from above to verify the shortest plan lengths. E.g., in a matter of a few minutes, the planner can find the shortest solution for our $n = 100$ instance with a length of 3,145,725 actions. In summary, the maturity of the technology produced by the planning community had a direct impact on our ability to iterate on ideas for the graph tracks quickly.

Listing 1: Single PDDL encoding using one move action.

```
(:action move
    :parameters (?l1 ?l2 - loc)
    :precondition (and
        ; Source has token
        (tokened ?l1)
        ; Destination is free
        (free ?l2)
        ; Destination's neighbors are free
        (forall (?l3 - loc) (imply
            (and (not (= ?l1 ?l3))
                (edge ?l2 ?l3))
            (free ?l3))))
    :effect (and
        ; Source is free
        (not (tokened ?l1)) (free ?l1)
        ; Destination has token
        (tokened ?l2) (not (free ?l2)))))
```

Listing 2: Split PDDL encoding using two actions.

```
(:action pick
    :parameters (?l1 - loc)
    :precondition (and
        ; Not holding a token
        (handfree)
        ; Source has token
        (tokened ?l1))
    :effect (and
        ; Holding a token
        (not (handfree)) (holding)
        ; Source is free
        (free ?l1) (not (tokened ?l1))))

(:action place
    :parameters (?l1 - loc)
    :precondition (and
        ; Holding a token
        (holding)
        ; Destination is free
        (free ?l1)
        ; Destination's neighbors are free
        (forall (?l2 - loc) (imply
            (edge ?l1 ?l2) (free ?l2))))
    :effect (and
        ; Not holding a token
        (not (holding)) (handfree)
        ; Destination has token
        (not (free ?l1)) (tokened ?l1)))
```

## 3.2 Other Graph Construction Algorithms

Despite our guarantee of exponential growth, there was one approach submitted by Bousquet, Durain, and Pierron (the *tpierron* team) that outperformed our construction. While we use our widget even for small graphs, the tpierron team brute-force the $n = 10$ case, focusing on the diameter of the graph generated by reconfiguration sequences.

For $n = 50$ and $n = 100$, the tpierron team uses a larger, more complex widget that has a longer reconfiguration sequence than ours. While we connect widgets of size 5, they connect widgets of size 10 in such a way that results in $4 \cdot d$ transitions in the previous graph $G$, where $d$ is the length of the reconfiguration sequence in the original graph $G$. That is, for every new widget added, the sequence is increased by a factor of 4 and 10 additional transitions are forced in the added widget to construct $G'$. Thus, they produce a sequence that grows at a rate of $4d + 10$.

They then connect widgets in a way that requires a complete sequence of transitions in the original graph for a partial sequence of transitions in the added widget. In contrast, we connect widgets in a way that requires a complete sequence of transitions between all widgets, i.e., each house widget must completely flip before another can be adjusted.

They do so while retaining the requirement that tokens can only move following the maximum reconfiguration sequence in both $G$ and $G'$. As a result, in the $n = 50$ scenario, the tpierron team achieved a reconfiguration sequence length of 3410 compared to our length of 3069, and in the $n = 100$ case, they reached a length of 3495250 compared to our length of 3145725. For more details on their submission, see "Graph track description" by Nicolas Bousquet, Bastien Durain, and Theo Pierron in the Core 2022 booklet (Soh, Okamoto, and Ito 2022).

## 4 Planning Encoding

The planning domain definition language (PDDL) is the de-facto standard language for modeling planning tasks (Haslum et al. 2019), and most planning tools are built with PDDL as their input language. The ISR problem can be encoded in PDDL by introducing a single lifted action to *move* a token from one location to another. Listing 1 shows the PDDL code for this *move* action, with comments interleaved. We call this the *single* encoding. While the encoding itself is quite compact, grounding these tasks is slow. In an ISR instance with $n$ nodes, $n^2$ *move* actions have to be created. As we are dealing with graphs of up to 40000 nodes, this can be problematic. To overcome this issue, we tested two approaches. The first is manual pre-grounding, called *single-grounded*. This does not help with the quadratic number of actions but avoids overhead creating the SAS$^+$ representation. The second approach, called *split*, is to split the move action into two actions, *pick* and *place*. It is presented in Listing 2. In this encoding, we only need $2n$ actions but plans are twice as long and have to be post-processed. Even this encoding can be slow to ground and can be sped up significantly with pre-grounding, which we call *split-grounded*. Ultimately, we found the *split-grounded* encoding to be the most efficient, and so we used it for all tracks and solvers.

The planning systems we used are all built on the Fast Downward planning system (Helmert 2006), which first translates the input PDDL into SAS$^+$ (Bäckström and Nebel 1995) before searching for a plan. While we used the aforementioned PDDL encodings for the bulk of the development work for the contest, our final submission directly encodes the input tasks into the *split* SAS$^+$ format to save on the computational effort required by this translation.

We encode a given ISR problem $\langle G, I_s, I_t \rangle$ with a graph $G = \langle V, E \rangle$, as an SAS$^+$ task $\langle \mathcal{V}, \mathcal{A}, \mathcal{I}, \mathcal{G} \rangle$ in the following way. The variables $\mathcal{V} = V \cup \{hand\}$ contain one binary

variable for each node in the graph to represent if there is a token on this node, and a binary variable *hand* to represent if we are currently holding a token. The domain of all variables is $\{free, occupied\}$. The initial state is $\mathcal{I} = \{v \mapsto occupied \mid v \in I_s\} \cup \{v \mapsto free \mid v \in V \setminus I_s\} \cup \{hand \mapsto free\}$, and the goal is $\mathcal{G} = \{v \mapsto occupied \mid v \in I_t\} \cup \{v \mapsto free \mid v \in V \setminus I_t\} \cup \{hand \mapsto free\}$. Note that specifying the occupied nodes in the goal would also be sufficient but specifying a value for all variables can help the planners realize that there is exactly one goal state.

The actions are analogous to the ones shown in Listing 2. There is an action $pick(v) \in \mathcal{A}$ for every $v \in V$ and it has the precondition $pre(pick(v)) = \{v \mapsto occupied, hand \mapsto free\}$ and effect $eff(pick(v)) = \{v \mapsto free, hand \mapsto occupied\}$. I.e., picking up a token is possible from all nodes that have a token, as long as we are not already holding one. Additionally, there is an action $place(v) \in \mathcal{A}$ for every $v \in V$ and it has the precondition $pre(place(v)) = \{v \mapsto free, hand \mapsto occupied\} \cup \{v' \mapsto free \mid \{v, v'\} \in E\}$ and effect $eff(place(v)) = \{v \mapsto occupied, hand \mapsto free\}$. So placing a held token is only possible on positions that currently have no token and have only free neighbors. The latter ensures every reachable configuration is an independent set.

## 5 Finding Solutions

We use sequential algorithm portfolios for each of the three solver tracks. That is, we run a sequence of algorithms, each with an associated time limit. The next section describes the algorithms that we use in our sequential portfolios.

### 5.1 Planning Algorithms

After testing various planning heuristics from the literature in exploratory experiments, we found that *landmark*-based heuristics to work well on ISR tasks. Relaxation-based heuristics, such as FF (Hoffmann and Nebel 2001) and Red-black (Domshlak, Hoffmann, and Katz 2015) did not contribute to search performance. Interestingly, both for satisficing and optimal planning, it is best to combine the landmark costs admissibly.

**A\*+Landmarks**   We run an A\* search (Hart, Nilsson, and Raphael 1968) with a *landmark count* heuristic (Karpas and Domshlak 2009) that uses two different kinds of landmarks: $h^1$ landmarks (Keyder, Richter, and Helmert 2010) and RHW landmarks (Richter, Helmert, and Westphal 2008). The landmark costs are combined with *uniform cost partitioning* (Katz and Domshlak 2008), which ensures that the resulting heuristic is admissible. As a result, this algorithm is optimal, sound, and complete, i.e., if it reports a plan, this is a shortest plan, if it reports unsolvability, the task is indeed unsolvable, and given sufficient resources, it will terminate.

**GBFS+Landmarks**   We run a greedy best-first search (GBFS) (Doran and Michie 1966) with a *landmark count* heuristic (Karpas and Domshlak 2009) over $h^1$ landmarks (Keyder, Richter, and Helmert 2010). Again, the landmark costs are combined with *uniform cost partitioning*. This algorithm is sound and complete, but not optimal.

**Symbolic Search**   We run a forward symbolic blind search (Torralba et al. 2017; Speck, Geißer, and Mattmüller 2020) using Binary Decision Diagrams (Bryant 1986) as the underlying data structure. The symbolic planner we use is SymK (Speck, Mattmüller, and Nebel 2020), which uses CUDD (Somenzi 2015) as its decision diagram library. This search is optimal, sound and complete.

**Symbolic Top-k Search**   The problem of finding a plan that is as long as possible is not commonly considered in the planning community, but only in the context of approximating the longest possible solution in SAT-based planning (Abdulaziz, Gretton, and Norrish 2017). Interestingly, the search for the longest path in a compactly represented graph is NEXPTIME-hard (Papadimitriou and Yannakakis 1986) and is therefore considered more complex than ordinary satisficing or optimal planning, which are known to be PSPACE-hard (Bylander 1994). Cohen, Stern, and Felner (2020) investigated heuristic search for finding the longest path for a given explicitly represented graph. While this is an interesting line of research to be applied in the context of planning, in the CoRe 2022 challenge we were interested in finding a long plan, but not necessarily the longest.

To find long plans, we run a forward symbolic blind search based on the algorithm SymK-LL (von Tschammer, Mattmüller, and Speck 2022), implemented in the symbolic search planner SymK (Speck, Mattmüller, and Nebel 2020), which iteratively finds and generates all loopless plans for a task. However, we have made the following adjustments to find long loopless plans. First, once we find a goal state reachable with cost $c$, we reconstruct only one loopless plan with cost $c$ and ignore all other plans with the same cost. Second, since the split encoding introduces intermediate states in which a token is held, we ignore these artificial states when evaluating if a plan is loopless during the plan reconstruction of SymK-LL. This algorithm iteratively finds longer plans, starting with the shortest one, and eventually finds the longest loopless plan, given enough resources.

**Counter Abstraction**   We abstract the problem to a planning problem that counts how many tokens are in certain positions and check for unsolvability in the abstraction. Since this algorithm is new, we describe it in more detail in Section 5.6. We now describe our sequential algorithm portfolios. Our portfolio for the *existent* track is identical to the one for the *shortest* track.

### 5.2 Portfolio for *shortest* and *existent* Tracks

The competition enforced no resource constraints and left it up to the competitors for how long they want to run their solvers. We decided on the following time limits for our portfolio based on some initial test that showed diminishing returns for higher limits. If one step in the portfoilo finds a solution, the remaining steps are skipped.

1. Counter abstraction: 10 seconds
2. Symbolic search: 70 minutes
3. A\*+Landmarks: 70 minutes
4. GBFS+Landmarks: 70 minutes
5. Counter abstraction: 14 hours

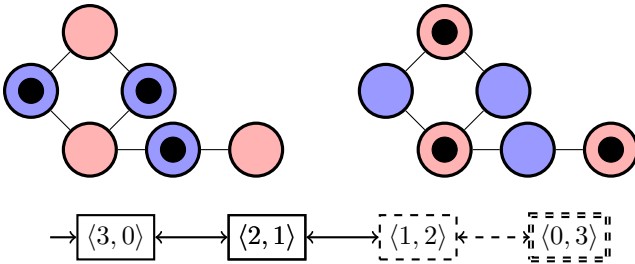

Figure 4: Example coloring for the counter abstraction approach. Top: initial state (left) and goal state (right). Nodes are colored blue if they have a token in the initial state but not the goal state and red if they have no token in the initial state but a token in the goal state. Bottom: the abstract state space. Dashed nodes are pruned.

Note that we use counter abstractions twice: first with a small time limit at the start of the portfolio to handle all cases where we can quickly prove unsolvability. Then again with a large time limit after all other components to catch unsolvable instances that are hard to prove unsolvable.

### 5.3 Single Solver for *shortest* and *existent* Tracks

We ran GBFS+Landmarks for 70 minutes as our single-solver submission because it has the highest coverage among all solvers in the portfolio.

### 5.4 Portfolio for *longest* Track

Our portfolio for the *longest* track ran two components: (1) GBFS+Landmarks: 330 seconds; and (2) Symbolic top-$k$ search: 70 minutes. When GBFS+Landmarks finds a solution, we use the cost of that solution as the lower solution bound for the subsequent symbolic top-$k$ search, so that only solutions that are longer than the solution we already have are reconstructed. As a fallback, if neither of the two approaches produced a solution longer than the shortest/existent tracks, we used the solution to the shortest/existent tracks as a default.

### 5.5 Single Solver for *longest* Track

We ran symbolic top-$k$ search for 70 minutes as our single-solver submission for the *longest* track. Note that we did not use the "fallback" option for this single-track submission.

### 5.6 Counter Abstraction

The *counter abstraction* component of our solver tries to detect if the task is unsolvable by abstracting it to a planning problem that counts the number of tokes in certain locations. This idea is inspired by counter abstractions in the area of model checking (e.g., Wahl and Donaldson 2010). Similar ideas where proposed in the area of planning as well (Riddle et al. 2016). In model checking, counter abstractions are usually used for symmetry reduction, whereas we do not require the abstracted parts to be symmetric to each other.

Given an ISR problem, we produce a *coloring* of the vertices in the graph, i.e., a function that maps each vertex of the graph to one color. Many different ways of coming up with

a good coloring are conceivable but we opted for a simple strategy that uses up to four colors: one each for nodes that

- contain a token both in the initial and in the goal state;
- contain a token only in the initial but not in the goal state;
- contain a token only in the goal but not in the initial state;
- are empty in the initial and goal state.

Colors for situations that do not occur are not used. For example, the task in Figure 4 only requires two colors.

Given a coloring, each original state can be abstracted to a state with one counter variable per color that tracks how many tokens currently are on vertices with this color. For example, the initial state in Figure 4 has 3 tokens on blue nodes and 0 on red nodes, so it can be represented as the state $\langle 3, 0 \rangle$. The goal has all three tokens on red nodes and none on blue, so it can be represented by $\langle 0, 3 \rangle$. Moving a token from a node colored $c_i$ to a node colored $c_j$ changes the abstract state from $\langle c_1, \ldots, c_i, \ldots, c_j, \ldots, c_n \rangle$ to $\langle c_1, \ldots, c_i - 1, \ldots, c_j + 1, \ldots, c_n \rangle$. The main observation is that if any solution to the full problem exists, there has to be a solution in the abstraction as well. We thus construct the state space of the abstraction in the following way.

For a state $s = \langle c_1, \ldots, c_n \rangle$, we construct one successor for each pair of unique colors $c_i$ and $c_j$ that differs from $s$ by a single token that moved from $c_i$ to $c_j$. In our running example, the initial state $\langle 3, 0 \rangle$ has a single successor $\langle 2, 1 \rangle$, and this state has two successors $\langle 3, 0 \rangle$ (which we skip because we have already seen this state) and $\langle 1, 2 \rangle$. The latter state now has the abstract goal $\langle 0, 3 \rangle$ as a successor (Figure 4).

Whenever we generate a state, we check whether such a state is possible (independent of whether it is reachable). If it is not possible to place the tokens on the respective colors in the required way, we do not have to consider it or its successors. In our running example, the state $\langle 1, 2 \rangle$ is not realizable: no matter where we place the blue token, it blocks two of the three red nodes. We use a mixed-integer program (MIP) solver to test if a state $s$ is realizable by checking if the following system of constraints has a solution:

$$x_i + x_j \leq 1 \quad \text{for all edges } \langle i, j \rangle \text{ in the graph}$$
$$\sum_{i \in N_c} x_i \geq s[c] \quad \text{for all colors } c$$
$$x_i \in \{0, 1\} \quad \text{for all nodes } i,$$

where $N_c$ is the set of all nodes with color $c$ and $s[c]$ is the amount of tokens that should have color $c$ in state $s$. The abstract state $s$ is realizable iff the constraints have a solution.

If we generate a state that matches the goal state ($\langle 0, 3 \rangle$ in our example), we know that there is an abstract plan. In this case, we still do not know if there is a real plan and return `unknown` (this component of the solver is incomplete). However, if there is no solution to the abstract problem, there cannot be a solution to the original problem. The abstract state space is usually small enough to explore completely. In our running example, it only has 4 states, and we only have to explore 3 of them, as we prune state $\langle 1, 2 \rangle$.

While the MIP we use to check for realizability of abstract states is specific to ISR, the rest of the technique is domain-independent, and we will explore this further in the future.

| | existent | | shortest | | | | longest | | | | CoRe 2022 limits | | |
|---|---|---|---|---|---|---|---|---|---|---|---|---|---|
| | #c | #e | #c | #e | c score | e score | #c | #e | c score | e score | time (s) | mem (GB) | cores |
| ReconfAIGERation | 257 | 246 | 152 | 214 | 201.36 | 214.00 | 54 | 29 | 83.02 | 29.00 | 10000 | 128 | 4 |
| junkawahara | 122 | 175 | 110 | 130 | 110.00 | 130.00 | 21 | 29 | 44.16 | 56.88 | 600 | 32 | 1 |
| PARIS | 334 | 322 | 275 | 275 | 282.74 | 280.12 | 143 | 233 | 183.24 | 251.53 | 62610 | 16 | 32* |
| telematik_tuhh | 326 | 303 | 280 | 267 | 280.00 | 267.00 | 27 | 32 | 76.51 | 87.95 | 144000 | 60 | 2 |
| toda5603 | 207 | 211 | 134 | 77 | 164.36 | 117.76 | 31 | 70 | 60.45 | 108.20 | $\sim$ 10000 | 32 | 1 |
| recongo | 244 | 240 | 238 | 236 | 238.00 | 236.00 | 115 | 26 | 155.93 | 26.00 | 12600 | 96 | 1 |

Table 1: Coverage results from both the competition (c) and our experiments (e). # indicates the total number of problems solved or found to be the shortest/longest. "score" refers to the IPC-style calculation over all of the problems (see text for further details). The last column reports the limits used by the teams in the competition. If different limits were used in different tracks, we report the maximum. Our solver mostly runs on 1 core but the MIP solver used by numerical abstractions used 32.

## 5.7 Other Competitors

Across all solver tracks, seven teams competed at CoRe 2022, including our team (PARIS). Three of them were classified as portfolios: our portfolio, the submission by Turau and Weyer (telematik_tuhh), and the one by Froleyks, Yu, and Biere (ReconfAIGERation) in the existent track.

The solver telematik_tuhh tackles the problem by searching in the space of independent sets with two algorithms running concurrently: an iterative deepening $A^*$ search using the number of misplaced tokens as heuristic value for finding optimal solutions, and a breadth-first search for detecting unsolvability. These algorithms are enhanced by domain-specific successor generation and memory optimization.

In the existent track ReconfAIGERation first transforms the problem to circuits represented as and-inverter graphs in the AIGER format (Biere, Heljanko, and Wieringa 2011), and then solves them with ABC (Brayton and Mishchenko 2010), a model checker that runs several algorithms concurrently. In the other tracks it represents tasks as SAT formulas encoding increasingly longer reconfiguration sequences. The resulting bounded model checking problems are solved by the incremental SAT solver CaDiCaL (Biere et al. 2020).

Among non-portfolio entries, the one by Yamada, Kato, Kosuge, Takeuchi, and Banbara (recongo) achieved strong results. They translate instances into answer set programs and leverage clingo (Gebser et al. 2019) as an off-the-shelf solver. Toda (toda5603) employs a modular strategy by initially running a greedy search and directly returning its suboptimal solution upon success. If it does not reach the goal, the problem is recast to a bounded model checking task where the state reached by the search is the initial state. This step is further informed by edge clique covers computed by ECC (Conte, Grossi, and Marino 2020) and solved by the bounded model checker NuSMV (Cimatti et al. 2002).

Kawahara and Yamazaki (junkawahara) work with families of independent sets, such as the initial independent set, or the family of all independent sets. They represent such families as zero-suppressed binary decision diagrams (ZDD; Minato 1993) and generate successors using set operations on ZDDs implemented using Graphillion (Inoue et al. 2016).

Lastly, Blé, Cui, Wu, and Zhong (tigrisg) rely on a state-action-reward-state-action approach. We refer to Soh, Okamoto, and Ito (2022) for the full solver descriptions.

## 6 Evaluation

In our experimental setup, we converted the docker images of each competing solver to singularity images (for improved performance) and ran all solvers in a unified setup with 2 hours timeout, 60 GB memory and 10 cores. All evaluations were run on Intel Xeon Silver 4114 processors running at 2.2 GHz. We could not include team tigris in the experiment since we could not run their docker container, and their team lost contact with the person who created it. In the PARIS portfolio we adjusted the resource allocation to distribute the time allocation to its components proportionally to the overall time limit rather than hard-coded, and fixed a bug (described below).

The shift in evaluation methodology is worth highlighting. In contrast with the contest parameters (where competitors were welcome to run their methods on their own hardware without any real resource constraints), we wanted to have a uniform analysis of the various approaches. This mitigates any bias that may stem from one team's computing infrastructure being more formidable than another. We can also see in the last columns of Table 1 that teams allocated very different amounts of resources to their solvers. By fixing one set of limits, we might bias the results towards a solver but we tried to select limits sufficiently high that all solvers can show their strengths.

Table 1 compares the coverage results we obtained with the ones from the competition. For the *existent* track, coverage dropped in most cases compared to the competition since the competition gave no restrictions on resource usage and most submissions had a significantly higher timeout. Teams junkawahara and toda5603 are the exceptions due to lower limits in the competition: the former used only 10 minutes and the latter only 32GB memory. However, the relative ordering of performance among the solvers remains.

For the *shortest* and *longest* tracks, solvers only gained coverage for a task if their solution was the best one amongst all competitors. This makes an analysis between the competition and our evaluation difficult since different best solutions might have been found. We note that for *shortest*, ReconfAIGERation shows improved performance, most likely because their submission used only 32GB of memory while we used 60GB. For *longest*, both ReconfAIGERation and recongo dropped significantly since they used a much higher

| | existent | | shortest | | longest | |
|---|---|---|---|---|---|---|
| | + | - | + | - | + | - |
| ReconfAIGERation | 70 | 1 | 56 | 0 | 201 | 1 |
| junkawahara | 181 | 10 | 150 | 10 | 215 | 14 |
| telematik_tuhh | 22 | 10 | 10 | 8 | 200 | 1 |
| toda5603 | 106 | 1 | 191 | 0 | 209 | 46 |
| recongo | 77 | 1 | 35 | 1 | 208 | 3 |

Table 2: Per-task comparison showing how often PARIS performed better (+) or worse (-).

time/memory limit in the competition; while PARIS performed significantly better. The latter is because in the competition, our submission for *longest* used the solutions from *shortest* as a seed to find longer plans, and we accidentally passed information that was processed incorrectly when we did not find a solution for *shortest*. For this experiment, we instead recomputed a (not necessarily shortest) plan in the beginning and handled the case of no found plan correctly.

We also included a scoring function that gives partial points for finding some solution; for *shortest* it is the ratio of the minimal reported solution and the found solution (analogous to the quality score used in IPC), for *longest* it is the ratio of the found solution and the maximal reported solution. The score suggests that in *shortest*, many solvers compute only minimal length solutions since their score is identical to their coverage. The picture is quite different for *longest*, showing that solvers that performed poorly often did find a decently long solution but not the longest overall.

Table 2 reports the number of tasks PARIS solved that others did not (+) and vice versa (-). Overall junkawahara is the most complimentary to our approach, with telematik_tuhh also solving some problems we could not on *shortest*, and toda5603 solving the most problems we did not on *longest*.

Figure 5 shows how many tasks each solver solves within a given time limit. In most cases, many problems are solved early on with more problems only trickling in slowly. The exceptions are the portfolio approaches, showing a sharp increase in coverage around the time the next component is started. A more sophisticated interleaving of the portfolio components could be used to smooth this process. We note that PARIS-l utilizes the full runtime on most problems. This is because it iteratively searches for longer plans, rarely terminating early, but producing longer and longer plans.

Finally, we reran each component of our *existent* portfolio separately to analyze their contribution. Mimicking our competition submission, we analyze how many tasks a component solves within 70 minutes and 16GB while none of the previous components could. Symbolic search solved 228 problems, A*+Landmarks added another 45, GBFS+Landmarks 16 more and finally the counter abstraction detected 37 problems as unsolvable. In the competition the higher timeout for this component lead to 9 more problems detected unsolvable, bringing the total coverage up to 335. We also compared how many tasks could only be solved by a single approach: 16 for symbolic search, 0 for A*+Landmarks, 16 for GBFS+Landmarks and 37 for the counter abstraction. While A*+Landmarks is dominated by

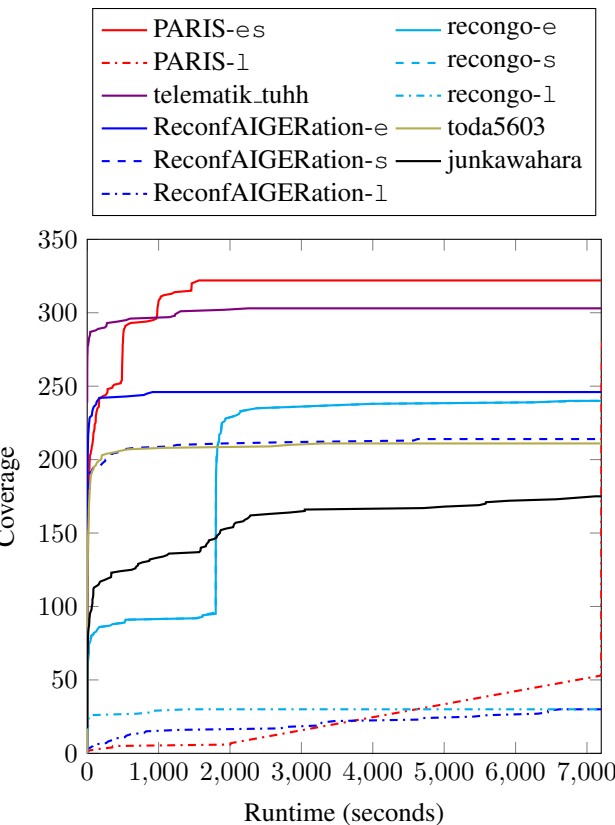

Figure 5: Number of tasks solved within a given time limit. If the same team had dedicated solvers for certain tracks, the track is indicated in the letter(s) behind the hyphen.

GBFS+Landmarks, it returns optimal solutions, making it an important contributor for the *shortest* track. The counter abstraction was invaluable for detecting unsolvabilty, as no other component could decide any of the tasks solved by it.

## 7 Conclusions

In this paper, we introduced the independent set reconfiguration problem, one of the most-studied problems of combinatorial reconfiguration, as a testbed for planning algorithms. We modeled this problem as a planning task and adapted different planning techniques for solving it, including a new technique for detecting unsolvable ISR problems that we think can be generalized to unsolvability planning in general. The resulting solvers participated successfully in the 1st Combinatorial Reconfiguration Challenge (2022), winning the majority of awards in multiple tracks. We reran all solvers of the competition under equal computational conditions for a more thorough analysis and investigated the strengths and weaknesses of our planning-based solvers. Our findings show that the independent set reconfiguration problem is an interesting and challenging problem for planning, and our algorithms are currently among the best approaches for solving it. We believe these findings will lead more planning researchers to develop ideas for the ISR problem and create a cross-fertilization of the fields.

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
