# OpenReview forum: "PARIS: Planning Algorithms for Reconfiguring Independent Sets"
_icaps-conference.org/ICAPS/2023/Workshop/HSDIP — ICAPS HSDIP 2023_

### Official Review · Reviewer_kYHG · 2023-04-25
**Review of PARIS: Planning Algorithms for Reconfiguring Independent Sets**

**Rating:** 9
**Confidence:** 3

**Review:**

The paper reviews the contribution of the PARIS planning system to the first combinatorial reconfiguration challenge (CORE). It provides a detailed description of the structure of the planning system, the challenge itself and presents a re-evluation of the challenge in a potentially fairer setting.  The experimental evaluation shows that PARIS is one of the top performers in the challenge, if not the best. Given the success of PARIS and so relevance in this context, the paper establishes a strong link between automated planning and combinatorial reconfiguration research.

The authors do an excellent job of conveying the content in an easily understandable manner for readers from the planning community. The experimental evaluation is thorough and well-conducted. As I am not familiar with the research area of combinatorial reconfiguration, it is difficult for me to fully determine the significance of this work. However, considering the diverse research backgrounds of the participants and our open attitude towards planning applications, this submission seems like a very high-quality contribution to me. I strongly recommend to accept this paper.

There is only one question on my mind, that I would like to pose for the open discussion: Are there any practical applications for combinatorial reconfiguration directly linked to a real world setting? Or is combinatorial reconfiguration primarily a theoretical concept with limited practical applications?

---

> ### Author Response · Authors · 2023-05-05
> **Response to Review**
>
> Thank you for the nice review!
>
> We are not experts in the research field concerned with combinatorial reconfiguration. However, there are several real world applications. For example, the description of a research project targeted at founding a common infrastructure for utilizing and applying the algorithmic technology of combinatorial reconfiguration (https://core.dais.is.tohoku.ac.jp/en/project/project_summary/) states the following:
>
> "Combinatorial reconfiguration also appears in industry. In particular, applications to “24/7 systems” such as power distribution systems are prominent. A power distribution network is designed as it may reduce the blackout duration when failure happens by supplying electricity via multiple numbers of routes. For example, the Japan standard benchmark model of power distribution networks has approximately ten octodecillion alternatives for the choice of network configurations. Even the computation of a single optimal network configuration among them is out of reach. Furthermore, even if we may compute a single optimal network configuration, we encounter another issue that is characteristic of 24/7 systems. Namely, upon a switching procedure to reconfigure the current configuration to the optimal one, we may not allow any power failure during the process. This requires us to develop a switching procedure that does not cause power failure upon the process over the state space consisting of approximately ten octodecillion alternatives from the current network configuration to an optimal network configuration. With this idea, the concept of combinatorial reconfiguration is indeed applied to power distribution systems."

---

### Official Review · Reviewer_vTf3 · 2023-04-26
**Good paper on applying planning to a new domain**

**Rating:** 8
**Confidence:** 4

**Review:**

## Summary

The authors introduce the problem of reconfiguring independent sets to the planning community, in part by providing several PDDL models. They describe their submissions to the 1st Combinatorial Reconfiguration Challenge based on portfolios of various existing planning algorithms and adaptations to achieve objectives uncommon in planning such as finding a long plan. In addition, the competition results are re-evaluated under the same computational conditions.


## General

The paper is well-written and structured.
It gives a very nice overview of how planning approaches can be used in a new domain.
They cover modeling the problem and finding a suitable combination of search
algorithms and heuristics. It is also very interesting to see how other encodings
and domain-dependent solvers perform compared to planning.

I appreciate that they not only describe the final result but also their
thought processes, intermediate steps, and non-fruitful approaches.
I agree, that libraries like PLANUTILS are essential for prototyping and
getting a feeling for which approaches might work best.
This shows also that planning approaches can be useful to explore the solution space
even if the final solution is not based on planning.

The section on planning coding illustrates very nicely how large the impact of an
optimized model can be so that grounding is feasible and that specific knowledge
is extractable by the planner.

The counter-abstraction to prove insolubility looks interesting, and I'm curious
to see how it can be generalized to be domain-independent.

The reevaluation under the same computational conditions is useful to make a
fairer comparison. I wonder why competition itself has not imposed restrictions.
The evaluation is thorough and gives nice insights into the impact of the
different portfolio parts.

It is nice to see that PARIS performs comparably well on the existing and shortest
tracks as telematik_tuhh, a domain-dependent A* and heuristic implementation.


Overall the paper gives a very nice overview of how planning approaches can
be applied to new domains and that the results are comparable to other encodings/
approaches or even outperform them. It also points out possible bottlenecks such
as grounding. Dealing with new problems is a good way to first develop
domain-dependent solutions that can then be generalized.

---

> ### Author Response · Authors · 2023-05-05
> **Response to Review**
>
> Thank you for the detailed and positive review!

---

### Decision · Program_Chairs · 2023-05-05

**Decision:**

Accept

**Comment:**

We are happy to inform you that your paper was accepted for presentation at the HSDIP workshop. The reviewers agree that this work is well done and also relevant and interesting to the workshop. Congratulations!

Feel free to adapt the paper based on the reviews where you deem it appropriate.